physiology, ecology

corticosterone, oxidative stress, glucocorticoid-induced oxidative stress, hormesis, bird

**Author for correspondence:**
Mark F. Haussmann
e-mail: mfh008@bucknell.edu

# Is there an oxidative cost of acute stress? Characterization, implication of glucocorticoids and modulation by prior stress experience

Ariana D. Majer[1], Vince J. Fasanello[1], Kailey Tindle[1], Brian J. Frenz[1], Alexis D. Ziur[1], Chelsea P. Fischer[1], Kelsey L. Fletcher[1], Olivia M. Seecof[1], Sarah Gronsky[1], Brian G. Vassallo[1], Wendy L. Reed[2], Ryan T. Paitz[3], Antoine Stier[4] and Mark F. Haussmann[1]

[1]Department of Biology, Bucknell University, Lewisburg, PA 17837, USA
[2]Department of Biology, University of Minnesota Duluth, Duluth, MN 55812, USA
[3]School of Biological Sciences, Illinois State University, Normal, IL 61790, USA
[4]Department of Biology, University of Turku, Turku, Finland

ID AS, 0000-0002-5445-5524; MFH, 0000-0003-0021-0561

Acute rises in glucocorticoid hormones allow individuals to adaptively respond to environmental challenges but may also have negative consequences, including oxidative stress. While the effects of chronic glucocorticoid exposure on oxidative stress have been well characterized, those of acute stress or glucocorticoid exposure have mostly been overlooked. We examined the relationship between acute stress exposure, glucocorticoids and oxidative stress in Japanese quail (*Coturnix japonica*). We (i) characterized the pattern of oxidative stress during an acute stressor in two phenotypically distinct breeds; (ii) determined whether corticosterone ingestion, in the absence of acute stress, increased oxidative stress, which we call glucocorticoid-induced oxidative stress (GiOS); and (iii) explored how prior experience to stressful events affected GiOS. Both breeds exhibited an increase in oxidative stress in response to an acute stressor. Importantly, in the absence of acute stress, ingesting corticosterone caused an acute rise in plasma corticosterone and oxidative stress. Lastly, birds exposed to no previous acute stress or numerous stressful events had high levels of GiOS in response to acute stress, while birds with moderate prior exposure did not. Together, these findings suggest that an acute stress response results in GiOS, but prior experience to stressors may modulate that oxidative cost.

## 1. Introduction

Organisms must balance competing demands in a mercurial environment. In vertebrates, this balance is coordinated by the release of glucocorticoids, which facilitate an endocrine stress response to environmental challenges [1]. Baseline glucocorticoids are linked to an individual's energetic state [2], and higher levels are indicative of frequent environmental challenges [3] and chronic stress [1]. In response to an acute stress, glucocorticoids quickly rise causing behavioural modifications and energy mobilization [4–6]. In the short term, glucocorticoids modulate their own release through negative feedback at the level of the hypothalamus and pituitary. Based on the specific challenge, this system offers flexibility to increase both the magnitude and duration of glucocorticoid release above baseline levels to stress-induced levels, and once the stressor is removed glucocorticoids recover back towards baseline [7]. Because of the extensive and variable effects of glucocorticoids on

vertebrate physiology, individual variation in the pattern of stress-induced glucocorticoid release and recovery often correlates with altered survival and reproduction [8].

While glucocorticoids play a key role in regulating cellular fuel levels, aerobic organisms also require oxygen to efficiently convert food into an ATP energy source. However, during this process, some oxygen molecules can form destructive reactive oxygen species (ROS). When ROS interact with biomolecules, they not only disrupt the function of these molecules but also create more reactive species, producing a chain reaction of damaging events that affect lipids, DNA and proteins [9]. While organisms possess mitigating antioxidant defences to help quench these ROS, the ensuing damage can overwhelm antioxidant defence capabilities and repair mechanisms, resulting in oxidative stress [9,10]. Though limited amounts of ROS serve important functions as regulators in signal processing, higher concentrations of ROS result in cumulative damage to biomolecules that can have adverse effects on an organism's health and longevity [9,11].

The role of oxidative stress in health and ageing is subject to extensive study, as are the processes that cause oxidative stress. Chronic exposure to stressful events and glucocorticoids may be one such generator of oxidative stress [12–14]. Recent findings suggest that acute stressors can also result in elevated oxidative stress levels [7,15], indicating that both chronic and acute exposure to glucocorticoids may result in increased levels of oxidative stress. We suggest to term this relationship between glucocorticoids and oxidative stress glucocorticoid-induced oxidative stress (GiOS). While multiple studies clearly suggest a link between chronic exposure to elevated glucocorticoids and oxidative stress [12–14], the possibility of GiOS during acute stress has not yet been explicitly tested. In addition, the acute stress response is a complex event involving diverse physiological systems, and thus a variety of physiological factors other than glucocorticoids may be responsible for the resultant increase in oxidative stress.

Regardless of the specific role of glucocorticoids in underlying increases in oxidative stress during an acute stress response, the prior experience of individuals to acute stressful events might also modulate how they are able to cope with oxidative stress in response to an acute stressor. Hormesis suggests that stressors that are considered harmful at high doses can have stimulatory or beneficial effects at low doses [16,17]. Such a biphasic relationship has been observed with factors such as nutrition, temperature, physical activity and the immune system [16,17]. In addition, relatively minor acute stressors appear to promote longevity and metabolic health, and upregulate antioxidant defences [17]. These effects appear to be even stronger when organisms are exposed to low-level stressors early in life [16,18]. Therefore, it is possible that small increases in glucocorticoids early in life could allow for the priming of the physiological defence processes involved in combating oxidative stress.

Here, we examined potential relationships between acute stress, glucocorticoids and oxidative stress using Japanese quail (*Coturnix japonica*) through three related experiments. First, we sought to characterize how an acute stress response affects oxidative stress, and if the oxidative stress response to an acute stressor may differ between phenotypically distinct organisms with different life histories using two breeds of Japanese quail. Second, because an acute stress response is a complex, multi-faceted, physiological process, we wanted to test whether corticosterone specifically was the main contributor to oxidative stress. While this has been largely assumed in the literature, a direct role of glucocorticoids on an acute rise in oxidative stress, or acute GiOS, has yet to be explicitly tested. Third, we wanted to characterize how prior experience with acutely stressful events may modulate acute GiOS.

## 2. Material and methods

### (a) Experiments

#### (i) Acute GiOS characterization in two breeds of Japanese quail

Ten domestic female Japanese quail (*Coturnix japonica*) of the Jumbo Brown breed were used from a colony maintained at Bucknell University. Jumbo Brown quail were selected for fast growth and meat production. Birds in our colony begin egg laying at six weeks of age, have adult masses that average $265 \pm 31$ g and an average lifespan of 3.0 years. The domestic quail were non-siblings and were age matched ($305 \pm 10$ days). Birds were housed in a single pen ($1.2 \times 1.2$ m) and maintained on a 12 L : 12 D cycle, with *ad libitum* access to food and water.

In addition, because domestic quail have undergone intense inbreeding, we also chose to study 10 feral female Japanese quail from a captive-bred colony maintained at North Dakota State University. The feral line was originally captured on the Big Island of Hawaii in 1980 and was maintained by the avian research centre at the University of British Columbia and therefore underwent less intensive selection than domestic breeds of *Coturnix japonica*. In 2008, 100 eggs were obtained from this population and used to begin a randomly breeding colony at North Dakota State University. Birds in our colony begin egg laying at 8–10 weeks of age, have adult masses that average $141 \pm 17$ g and an average lifespan of 5.0 years. These 10 feral quail were non-siblings, age matched ($365 \pm 21$ days), housed in a single pen with a breeding colony that was at the same density as that of the domestic quail ($1.2 \times 2.7$ m) and maintained on a 12 L : 12 D cycle with *ad libitum* access to food and water. Both lines of quail were exposed to a single acute stress series via bag restraint protocol (described below).

#### (ii) Causal involvement of glucocorticoids in acute GiOS

For two weeks prior to the experiment, 32 domestic quail ($230 \pm 15$ days of age) were housed in groups of four in pens ($1.2 \times 1.2$ m). Each pen was modified by placing an opaque white plastic divider that separated the pen into two equal halves, with a pair of quail on each side of the divider. Quail were kept in pairs so that individuals did not experience any isolation stress during this experiment. *Ad libitum* access to food and water was available on each side of the pen. In addition, one metal plate was placed on each side of the pen that was covered by a metal cover attached to a line that ran up and out of the pen and into a different room. Each morning, when the pens were entered to provide fresh food and water, a mealworm injected with sesame oil was placed on each metal plate under the cover. Within the next 2 h, the lines connected to the covers were pulled exposing the mealworm to the quail pairs in each half of each pen. One of the two quails in each pair would quickly eat the mealworm, though the specific quail within the pair that ate the worm was variable each day. The identity of the bird eating the worm and the precise time were observed through a window from the adjacent room. Direct observations and videotaping of trials did not reveal any dominance behaviours that appeared to influence feeding behaviours of these quail.

On the morning of the experiment, fresh food and water was supplied as normal, but on one side of the pen, we placed a mealworm injected with corticosterone in sesame oil (0.005 mg) and on the other side we placed a mealworm injected with sesame oil as a control. Quail were monitored through the window from the adjoining room, and once the metal cover was raised, we noted which bird in the pair ate the worm and began a timer. Blood samples were taken from the bird that ate the worm at either 10 min post-ingestion ($n = 10$; 5 control and 5 corticosterone-fed) or 45 min post-ingestion ($n = 22$; 11 control and 11 corticosterone-fed). After the first day, 16 of the 32 birds had eaten a worm and been bled. This protocol was repeated over the next 3 days until the other 16 quail ate a worm.

### (iii) Effects of prior stress experience on acute GiOS

We randomly assigned 30 domestic quail ($30 \pm 4$ days of age) into three treatment groups: high-stress frequency, low-stress frequency and naive ($n = 10$ per group). Over a period of 24 days, the birds were housed in pens ($1.2 \times 1.2$ m) and each group was exposed to a different number of acute stressors and on the final day of treatment, all birds from all groups were subjected to an acute stress test in which blood samples were taken from each bird. The high-frequency group was subjected to eight acute stress tests over the period or approximately one acute stress test every 3 days. The low-frequency group was subjected to two acute stress tests (day 1 and day 24). The naive group was subjected to one acute stress test on day 24. While the acute stressor was administered multiple times, blood samples were only taken at 53 days of age (day 24) to determine how prior exposure to acute stress affects the acute stress response and oxidative stress.

## (b) Procedures
### (i) Acute stress series

At the start of the light cycle (07.00), we collected a baseline sample of approximately 75 µl of blood from the alar vein within 3 min of entry into the room. Following baseline sample collection, we placed the bird into a breathable cotton bag. We collected a second sample, the stress-induced sample, at 15 min from entry into the room. Following this sampling period, we returned the birds to their home pens and left the room. We collected a third sample at 45 min after the initial room entry, and this sample served as a recovery sample, which was estimated to be during the time that negative feedback was turning off the acute stress response. Blood samples were collected into heparinized capillary tubes, transferred to microcentrifuge tubes and kept on ice for a short period until refrigerated centrifugation (4°C) for 5 min at 3000 rpm. Plasma was removed and stored at −20°C.

### (ii) Corticosterone RIA

We determined corticosterone levels by radioimmunoassay (RIA) based on the protocol of Wingfield & Farner [19] and validated in our laboratory [7,20]. Briefly, we extracted corticosterone from diluted plasma samples with anhydrous diethyl ether, dried under nitrogen gas, resuspended in 90% ethanol and stored at 4°C overnight. Samples were then centrifuged, and the supernatant was dried under nitrogen gas and resuspended in phosphate-buffered saline with gelatin (PBSg). We ran the samples in a competitive-binding RIA using corticosterone-specific antibody corticosterone-3-carboxymethyl-oxime: BSAhost: rabbit (MP Biomedicals, Solon, OH, USA) and tritiated corticosterone (2000 cpm, NET 399; New England Nuclear Research Products, Boston, MA, USA). Bound and free corticosterones were separated using dextran-coated charcoal, and after centrifugation, radioactivity was measured using a liquid scintillation counter. All samples were run in duplicate, which were averaged for analysis, and the intra-assay and inter-assay coefficients of variation were 7.1% and 7.4%, respectively.

### (iii) Oxidative stress measures

We assessed reactive oxygen metabolites (ROMs) and total antioxidant capacity (TAC), using the D-ROMs and OXY-adsorbent tests (Diacron International, Grosseto, Italy), respectively, in the baseline and 45 min blood samples as previously described [7] and validated [21]. Because of time and resource limitations, we could only choose two of the three blood samples for each bird that underwent an acute stress test to sample oxidative stress measures. In the previous work, we had determined that in this situation using the baseline (less than 3 min) and recovery (45 min) samples was most appropriate, as oxidative stress can change over this duration of an acute stress response [7].

The d-ROMs test measures intermediate oxidative damage compounds, mostly represented by hydroperoxides, derived from lipids, proteins and nucleotides. Specifically, we diluted 10 µl of plasma in 200 µl of the provided acidic buffer solution and then read the plate kinetically (one read per min) for the next 10 min. Absorbance was measured at 490 nm (BioTek ELx800, VT, USA) and we calculated change in ROMs concentrations (in mM of $H_2O_2$ equivalents) from these absorbencies by taking the difference between the reading at min 20 and min 0, dividing by 20 min and multiplying by the constant 9000 (all per the manufacturers specifications). All samples were run in duplicate, which were averaged for analysis, and the intra-assay and inter-assay coefficients of variation were 4.1% and 5.2%, respectively.

The total plasma antioxidant capacity (TAC) was measured using the OXY-Adsorbent test (Diacron International, Grosseto, Italy), which measures the effectiveness of the plasma antioxidant barrier by quantifying its ability to cope with oxidant action of hypochlorous acid (HClO). We diluted 10 µl plasma in 990 µl of distilled water; we then mixed 5 µl of this diluted plasma with 195 µl of the provided HClO solution and continued by following the manufacturer's instructions. We measured the absorbance at 490 nm (BioTek ELx800) and we calculated TAC (in mM of HClO neutralized). All samples were run in duplicate which were averaged for analysis and the intra-assay and inter-assay coefficients of variation were 6.2% and 4.1%, respectively.

## (c) Statistics

We ran statistical analyses using JMP Pro software (v.14.0.0, SAS Institute Inc. 2018, Cary, NC, USA). For all analyses, we performed generalized linear mixed models of restricted maximum likelihood (REML-GLMM). For every model, we checked for homogeneity of variances (Levene's test) and for normality of residuals (Kolmogorov–Smirnov test). In each model, we introduced 'individual' as a random factor to control for the non-independence of data due to repeated measurements on the same individuals. We sequentially removed non-significant interactions from the models, starting from the higher-order interactions, and repeated the analyses until we obtained a model with only significant terms. We carried out post hoc comparisons using Tukey HSD tests.

### (i) Experiment A (acute GiOS in two breeds)

For corticosterone analyses, breed, time (three time points) and the breed × time interaction were included as fixed effects. For oxidative stress analyses, ROMs and TAC were each modelled over the acute stress period (two time points). Oxidative stress models included the fixed effects of time, breed and the interaction of breed × time.

### (ii) Experiment B (role of glucocorticoids in acute GiOS)

For plasma corticosterone analyses, treatment, time (two time points) and the treatment × time interaction were included as fixed effects. For oxidative stress analyses (ROMs and TAC), treatment was included as a fixed effect.

### (iii) Experiment C (effects of prior stress experience on acute GiOS)

For corticosterone analyses, stress exposure treatment (three treatments), time (three time points) and the treatment × time interaction were included as fixed effects. For oxidative stress analyses, ROMs and TAC were each modelled over the acute stress period (two time points). Oxidative stress models included the fixed effects of time, stress exposure treatment and the interaction of treatment × time.

## 3. Results

### (a) Experiment A: characterization of acute GiOS in two breeds

In both feral and domestic quail breeds, corticosterone levels changed over the acute stress period (figure 1a, table 1). Specifically, corticosterone levels increased between the baseline and 15 min sample and decreased back to baseline levels by the 45 min sample (baseline–T15: $p = 0.007$; baseline–T45: $p = 0.753$; T15–T45: $p = 0.039$; figure 1a). Though the effect of breed on corticosterone levels did not reach statistical significance ($p = 0.055$; table 1), it is important to note that while 9 of the 10 domestic birds did show an increase in corticosterone between 0 and 15 min, many of those increases were modest (figure 1) compared with the feral quail. While there was also no effect of breed on ROMs or TAC levels (table 1), interestingly ROMs levels increased over the stress response (figure 1b) while TAC levels decreased (figure 1c; table 1).

### (b) Experiment B: causal involvement of glucocorticoids in acute GiOS

For quail that ingested a mealworm injected with corticosterone, there was a significant treatment × time interaction for corticosterone levels (figure 2a, table 2). Compared with control quail (fed sesame oil), corticosterone-fed quail had significantly higher plasma corticosterone levels at 10 min post-ingestion ($p < 0.0001$), but plasma corticosterone levels in corticosterone-fed birds returned to control quail levels by 45 min post-ingestion ($p = 0.814$; figure 3a). At 45 min post-ingestion, ROMs were significantly increased in the corticosterone-fed quail but not the control quail (figure 2b, table 2), while TAC was not significantly affected in either the control or the corticosterone-fed quail (table 2).

### (c) Experiment C: effects of prior stress experience on acute GiOS

Regardless of prior stress treatment, corticosterone levels changed over the acute stress period (figure 3a, table 3). Specifically, corticosterone levels increased between the baseline and 15 min sample, and then decreased between the 15 and 45 min sample, though the 45 min sample did not decrease to baseline levels (baseline–T15: $p < 0.0001$; baseline–T45: $p = 0.023$; T15–T45: $p = 0.012$; figure 3a). Interestingly, there was a

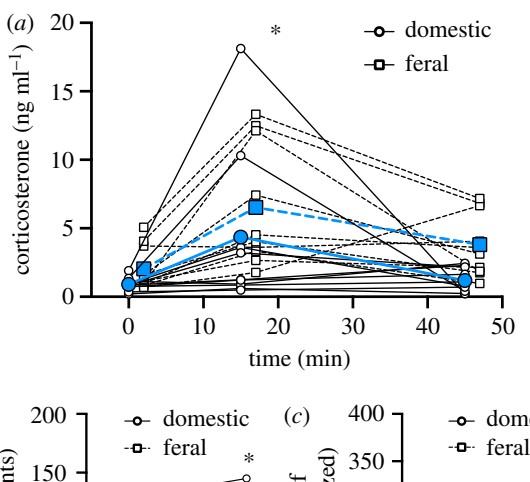

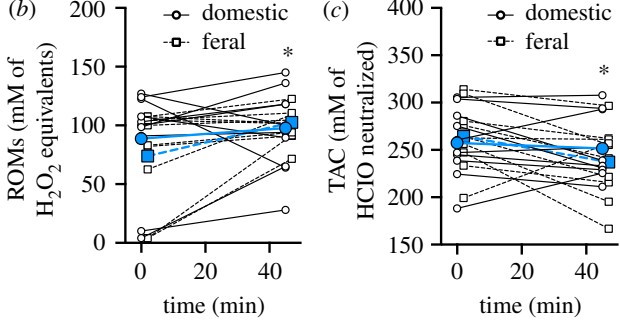

**Figure 1.** Acute bag restraint stress affects corticosterone and oxidative stress in Japanese quail. The effect of a domestic quail breed (open circles and solid lines) or a feral quail breed (open squares and dashed lines) on (a) plasma corticosterone levels, (b) reactive oxygen metabolites and (c) total antioxidant capacity. Scatter plots are shown, and to display individual variation the data points from individuals are joined by lines. Group means are denoted by larger filled symbols and lines. An asterisk denotes statistically significant difference among times ($p < 0.05$). For more statistical information, see table 1 and Results. (Online version in colour.)

significant treatment × time interaction for ROM levels (figure 3b; table 3). While ROM levels did not differ at baseline ($p > 0.977$), the naive and high-frequency stress groups showed an increase in ROMs between baseline and 45 min, while the low-frequency stress group had a decrease in ROMs between baseline and 45 min (naive: $p = 0.045$; low: $p = 0.001$; high: $p = 0.048$; figure 3b, table 3). The result was that by 45 min after stress initiation, the naive and high-frequency stress groups had similar levels of ROMs, but both had higher levels than the low-frequency stress group (naive–high: $p = 1.000$; naive–low: $p = 0.002$; high–low: $p = 0.002$; figure 3b, table 3). There was no effect of treatment, time or a treatment × time interaction on TAC levels.

## 4. Discussion

Our study demonstrates that during an acute stress response, individuals experience an increase in oxidative stress. We observed this increase in two different breeds of Japanese quail with distinct phenotypes, suggesting that an increase in oxidative stress during an acute stress response may be a general consequence of mounting an acute stress response. In addition, when quail ingested corticosterone in the absence of an acute stressor, oxidative damage levels increased. This suggests that glucocorticoids are at least partially responsible for the acute stress effect on oxidative stress, and thus we term this event acute GiOS. Prior experience of stressful events also affected the magnitude of acute GiOS because while both a novel stressor and a frequently repeated stressor

**Table 1.** Results of generalized linear mixed models on the response to breed (domestic or feral quail), time period of sample (0, 15 and 45 min for corticosterone and 0 and 45 min for oxidative stress measures) and their interaction. For corticosterone, ROMs and TAC $n = 20$ quail. Italic values indicate $p < 0.05$.

| dependent variable | breed | time | breed × time |
|---|---|---|---|
| corticosterone | $F_{1,18} = 4.211$, $p = 0.055$ | $F_{2,36} = 9.038$, $p < 0.0001$ | $F_{2,36} = 0.319$, $p = 0.728$ |
| ROMs | $F_{1,18} = 0.133$, $p = 0.7191$ | $F_{1,18} = 5.581$, $p = 0.029$ | $F_{1,18} = 1.524$, $p = 0.233$ |
| TAC | $F_{1,18} = 0.071$, $p = 0.7923$ | $F_{1,18} = 4.854$, $p = 0.041$ | $F_{1,18} = 2.025$, $p = 0.172$ |

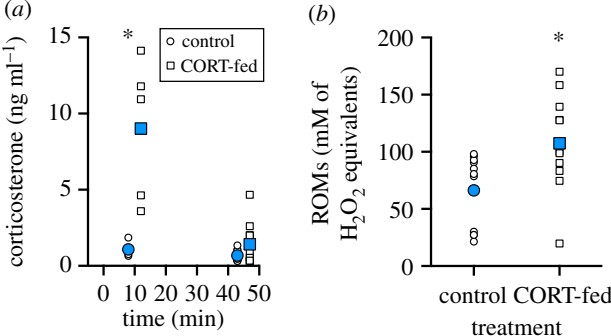

**Figure 2.** The ingestion of sesame oil (control) or corticosterone in sesame oil (corticosterone-fed) on (*a*) plasma corticosterone levels at 10 and 45 min post-ingestion, and (*b*) reactive oxygen metabolites at 45 min post-ingestion. Scatter plots are shown with points jittered to display individual variation and group means are denoted by larger blue symbols and lines. An asterisk denotes statistically significant difference among groups ($p < 0.05$). For more statistical information, see table 2 and the results. (Online version in colour.)

appear to induce acute GiOS, it seems that low levels of a repeated acute stressor may actually reduce acute GiOS.

A typical acute stress response of glucocorticoids, with a rapid rise to stress-induced levels followed by a recovery of baseline glucocorticoid levels upon the termination of the stressor, is shared across diverse vertebrate taxa [22]. While this acute glucocorticoid stress response is conserved, the degree to which GiOS is conserved across taxa remains to be determined. Here, we focused on differences in acute GiOS by studying two genetically, morphologically, physiologically and behaviourally distinct breeds within the same species of Japanese quail [23,24]. We did not see a statistically significant effect of breed on corticosterone, ROMs or TAC suggesting that while these two breeds differ in phenotype, their GiOS patterns could be similar. It is important to note that in our limited sample size, the change in corticosterone across the acute stress response in the domestic quail was relatively modest compared with the feral quail. Previous studies have found that domesticated species tend to have lower corticosterone levels [25,26] and HPA activity [26,27] as compared with wild species. Even so, both breeds of quail showed a significant change in GiOS, though again, the domestic quail GiOS responses were less robust than the feral quail responses. While GiOS itself has not been studied in much detail, other studies in domestic chickens (*Gallus gallus*) and prairie voles (*Microtus ochrogaster*) have also demonstrated increased oxidative stress in response to

an acute stressor in the form of increased levels of oxidative damage [7,20,28]. However, studies in wild king penguins (*Aptenodytes patagonicus* [29]) and a captive population of house sparrows (*Passer domesticus* [30]) saw either no significant change or even a decrease in oxidative damage levels in response to acute stress. We therefore need more research across diverse vertebrate taxa to determine the shared nature of the stress response and oxidative stress in order to better understand the extent to which GiOS is conserved.

Oxidative stress sustained during an acute stress response could be a result of many underlying factors, one of which is the exposure to elevated glucocorticoid levels. While previous studies have demonstrated a clear link between prolonged exposure to high glucocorticoid concentrations and increased levels of oxidative stress [12,14], the link between natural increases in glucocorticoids through acute stress and oxidative stress (or acute GiOS) is more tenuous. More specifically, the effect of acute glucocorticoid rises on oxidative stress in the absence of an external stressor has never been demonstrated. Previous studies reporting changes in oxidative stress levels in response to acute stress assumed that glucocorticoids were the mediators of such effects without any definitive evidence [7,15,29]. Here, we successfully managed to increase corticosterone levels acutely in the absence of external stress (i.e. not manipulating the birds or disturbing them by entering the room) through the ingestion of corticosterone. We showed that after 45 min, while corticosterone levels returned to baseline levels in birds that had ingested corticosterone, there were significantly higher levels of ROMs compared to birds who had not consumed corticosterone. These data suggest that elevated glucocorticoid levels are probably involved in the mechanism underlying increases in oxidative stress in response to an acute stressor. Future studies could also consider blocking glucocorticoid release to determine if the absence of a rise in corticosterone during acute stress is sufficient to prevent any concomitant rise in oxidative stress levels.

While our study shows that acute rises in glucocorticoids and shifts in oxidative stress are linked, we did not specifically explore how glucocorticoids might result in GiOS. Since mitochondria, which are the main sources of cellular ROS [31], possess glucocorticoid receptors [32], one possibility is that during acute stress, mitochondrial oxygen consumption and total energy expenditure increase, leading to a higher production of ROS [33]. However, these effects may be taxon-dependent, since a recent study in lizards showed that exogenous corticosterone supplementation was

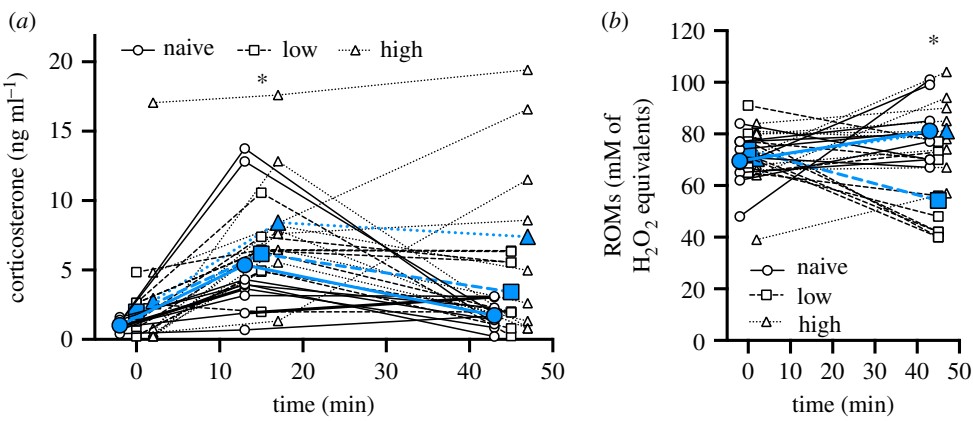

**Figure 3.** No prior acute restraint stress exposure (naive), low levels of prior acute restraint stress exposure (low) or high levels of prior acute restraint stress exposure (high) on (*a*) plasma corticosterone levels, and (*b*) plasma reactive oxygen metabolites during an acute bag restraint stressor. Scatter plots are shown with points jittered and to display individual variation, the data points from individuals are joined by lines. Group means are denoted by larger blue symbols and lines. An asterisk denotes statistically significant difference among (*a*) times or (*b*) groups ($p < 0.05$). For more statistical information, see table 3 and the results. (Online version in colour.)

**Table 2.** Results of generalized linear mixed models on the response to mealworm treatment (control or corticosterone-fed), time period of sample (10 and 45 min) and their interaction. For corticosterone, $n = 10$ at 10 min and $n = 22$ at 45 min, for ROMs and TAC $n = 22$ quail at 45 min. Italic values indicate $p < 0.05$.

| dependent variable | treatment | time | treatment × time |
|---|---|---|---|
| corticosterone | $F_{1,28} = 34.041$, $p < 0.0001$ | $F_{1,28} = 28.869$, $p < 0.0001$ | $F_{1,28} = 23.500$, $p < 0.0001$ |
| ROMs | $F_{1,20} = 6.827$, $p = 0.0167$ | | |
| TAC | $F_{1,20} = 2.046$, $p = 0.1680$ | | |

actually able to decrease mitochondrial ROS production [34]. The effect of glucocorticoids on mitochondria is still under extensive research and may be just one of several mechanisms behind how glucocorticoids could cause a shift into a state of oxidative stress.

Interestingly, the extent to which individuals are sensitive to GiOS could depend on their prior experience with acute stressors. The theory of hormesis suggests that low doses of stressors may actually be beneficial for an individual's health and longevity by improving metabolic health and upregulating antioxidant activity [16,17]. In the context of GiOS, this could mean that low doses of stress could alter the amount of oxidative stress experienced by an individual in response to a stressor later in life. Accordingly, in response to varying the frequency of exposure to acute stressors prior to measuring acute GiOS, we found that both the naive birds and the high-frequency-stress birds experienced an increase in ROMs over the stress response series, which, in agreement with the prior work in our laboratory, suggests that the rise in glucocorticoids during an acute stress response shifts an individual into a state of oxidative stress [7]. On the contrary, the low-frequency stress birds experienced a rapid decrease in ROMs over the acute stress test. This suggests that, in the low-frequency group, prior exposure to glucocorticoids may have primed the system to cope with the concomitant increase in ROS production by upregulating antioxidant defences and repair mechanisms [18]. A recent study in

king penguins found that the glutathione antioxidant system was upregulated both in birds with higher baseline corticosterone levels and following an acute restraint stress [29]. It is therefore possible that, while prior mild acute stress exposure has no effect on TAC, it may prime endogenous antioxidant defences like glutathione. Additionally, the birds used in this experiment were young, and other studies have supported the idea that hormetic effects are stronger when exposure to stressors occurs early in life. For example, a study in zebra finches demonstrated a significant decrease in oxidative damage in response to heat stress in adult birds exposed to heat stress early in life as compared with adult birds exposed to heat stress only in adulthood or never exposed to heat stress at all [18]. These observations in combination with our findings that low levels of acute stress early in life can result in lower levels of GiOS suggest that there is a hormetic response induced by a conditioning exposure to glucocorticoids that has the ability to prime an organism to better respond to stress and to better combat oxidative stress later in life.

## 5. Conclusion: GiOS in an ecological and evolutionary framework

Glucocorticoids and the information they provide on vertebrate responses to ever-changing environmental challenges

**Table 3.** Results of generalized linear mixed models on the response to prior acute stress experience (naive, low stress or high stress), time period of sample (0, 15 and 45 min for corticosterone and 0 and 45 min for oxidative stress measures) and their interaction. For corticosterone, ROMs and TAC $n = 27$ quail. Italic values indicate $p < 0.05$.

| dependent variable | treatment | time | treatment × time |
|---|---|---|---|
| corticosterone | $F_{2,24} = 3.076$, $p = 0.0647$ | $F_{2,24} = 16.477$, $p < 0.0001$ | $F_{4,48} = 1.117$, $p = 0.360$ |
| ROMs | $F_{2,24} = 4.280$, $p = 0.0257$ | $F_{1,24} = 0.073$, $p = 0.7893$ | $F_{2,24} = 10.597$, $p = 0.0005$ |
| TAC | $F_{2,24} = 2.048$, $p = 0.1509$ | $F_{1,24} = 1.217$, $p = 0.2808$ | $F_{2,24} = 1.788$, $p = 0.1889$ |

have become a mainstay measure in physiological ecology [3,35]. More recently, oxidative stress has also become a valuable tool to physiological ecologists, since the ability of an organism to manage oxidative stress is likely to be a major determinant of life histories [9]. Here, we detail how these two important physiological measures may be closely linked, as acute increases in glucocorticoids also cause increases in oxidative stress. While multiple studies have assumed a causal relationship between glucocorticoids and changes in oxidative stress levels in response to an acute stressor, no prior studies have demonstrated that glucocorticoids are in fact responsible. Our results strongly suggest that glucocorticoids are an underlying mechanism behind the increase in oxidative stress experienced by individuals in response to an acute stressor. Furthermore, we report that the degree to which an individual experiences GiOS is influenced by their previous exposure to acute stress. While previous stress exposure is more easily studied in the laboratory, where a controlled setting and continual animal monitoring allows us to know prior stress exposure, natural environments are inherently more realistic with more variable stressors. Thus, while this work lays the foundation for exploring these relationships, future field studies are needed to begin to fully describe how prior stressful experiences may alter these responses.

However, there is still much we do not know about GiOS, such as whether or not the oxidative damage produced is transient, like the glucocorticoids which appear to cause it, or if the oxidative damage accumulates over time. If it is the former, short-term increases in oxidative stress may simply act as a cellular signalling pathway to provide the cell with information on the physiological challenge happening globally throughout the organism. If it is the latter, then oxidative stress may be a cost that is paid against the obvious adaptive advantages provided by an acute rise in glucocorticoids. However, these are not mutually exclusive. Given that other work in physiological ecology has linked elevated glucocorticoids with telomere shortening, a marker of biological ageing [7,20,36–41], it is possible that GiOS acts as a mechanism by which stress exerts its detrimental effects on the ageing process. Future work should begin to tease apart the potentially complex roles of GiOS and explore whether long-term costs exist that may have effects on organismal fitness.

Ethics. All procedures were conducted with approval from the Bucknell University Institutional Animal Care and Use Committee.

Data accessibility. The datasets supporting this article have been uploaded as part of the electronic supplementary material.

Authors' contributions. M.F.H., W.L.R. and V.J.F. conceived of and designed the study. M.F.H., V.J.F., K.T., C.P.F., K.L.F., O.M.S. and S.G. conducted the animal work, and V.J.F., K.T. and C.P.F. did the laboratory work. M.F.H. performed statistical analysis. A.D.M., M.F.H., K.T., B.J.F., A.D.Z., A.S., R.T.P. and W.L.R. drafted and edited the manuscript.

Competing interests. We declare we have no competing interests.

Funding. This work was supported by the National Institutes of Health (grant no. 1R15-HD083870-01A1) to M.F.H. and R.T.P., and the Bucknell Biology Department.

Acknowledgements. We thank Cindy Rhone, Amber Hackenberg and Gretchen Long for help with the quail. We also thank Brad Eames and Shannon McCollum for helpful discussion of the manuscript.

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
