## [Reviewer comments · Proceedings of the Royal Society B: Biological Sciences]

Review History

RSPB-2019-1698.R0 (Original submission)

Review form: Reviewer 1

Recommendation

Accept with minor revision (please list in comments)

Scientific importance: Is the manuscript an original and important contribution to its field?

Excellent

General interest: Is the paper of sufficient general interest?

Good

Quality of the paper: Is the overall quality of the paper suitable?

Excellent

Is the length of the paper justified?

Yes

Should the paper be seen by a specialist statistical reviewer?

No

Do you have any concerns about statistical analyses in this paper? If so, please specify them explicitly in your report.

No

It is a condition of publication that authors make their supporting data, code and materials available - either as supplementary material or hosted in an external repository. Please rate, if applicable, the supporting data on the following criteria.

Is it accessible?

Yes

Is it clear?

Yes

Is it adequate?

Yes

Do you have any ethical concerns with this paper?

No

Comments to the Author

I thoroughly enjoyed reading this manuscript, which presents the most comprehensive test of the relationship between acute stress/glucocorticoids and oxidative stress to date. As such it provides an important and widely relevant advance on a topic that has received much speculation, but on which published data are relatively rare. I found it to be well-written and concise, and the findings are clear. I expect it to be of broad general interest.

My suggestions primarily consist of recommendations to better emphasize and provide context for some of the key findings.

First, the finding that moderate levels of prior stress decrease the ROMs response to subsequent stressors – but not the CORT response – is very interesting. These patterns, as well as the implication that moderate stressors may have hormetic effects on subsequent damage resulting from stressors, were well-described. But there wasn't much reference to the apparently paradoxical finding while CORT increases ROMs, moderate stress doesn't reduce the CORT stress response, or alter antioxidant capacity (at least as measured via TAC), but it does reduce the resulting oxidative damage. Presumably this is because some unmeasured components of antioxidant capacity or other physiological mediators increase to combat the direct effects of CORT on oxidative damage. The possibility of upregulating antioxidant defenses is mentioned briefly on lines 328-331, but the contrast between this and the finding of no change in TAC following repeated stressor exposure isn't explicitly discussed. Further discussion of what this might constitute given the lack of a response in measured antioxidants would strengthen the manuscript.

Similarly, noting that these results show that despite clear causal relationships between CORT and ROMs higher stress-induced CORT doesn't always equate to higher ROMs (as shown in the

prior stressor experiment) would be helpful. Along those lines, can you say anything about why previous studies might not have found an increase in ROMs under acute stress?

The experimental design is elegant and well-justified, and it's clear that the birds need to be housed in pairs. Nevertheless I found myself wondering whether there was any relationship between dominance status in the quail pairs and the propensity to consume mealworms. It seems like the design of the experiment would minimize the risk that any such differences would affect the findings, since both animals ultimately consumed a mealworm in the actual experiment. But if there were differences in the perceived quality of CORT and control-injected mealworms this could still matter. Providing a very brief mention of whether there was any indication that individuals discriminated between mealworm types, and/or whether there was a relationship between dominance/the propensity to consume mealworms during the training phase and the type of worm that individuals consumed during the experimental phase would address this.

Other suggestions:

- Remove the word "them" in line 164

Review form: Reviewer 2

Recommendation

Accept with minor revision (please list in comments)

Scientific importance: Is the manuscript an original and important contribution to its field?

Good

General interest: Is the paper of sufficient general interest?

Excellent

Quality of the paper: Is the overall quality of the paper suitable?

Good

Is the length of the paper justified?

Yes

Should the paper be seen by a specialist statistical reviewer?

No

Do you have any concerns about statistical analyses in this paper? If so, please specify them explicitly in your report.

No

It is a condition of publication that authors make their supporting data, code and materials available - either as supplementary material or hosted in an external repository. Please rate, if applicable, the supporting data on the following criteria.

Is it accessible?

N/A

Is it clear?

N/A

Is it adequate?

N/A

Do you have any ethical concerns with this paper?

No

Comments to the Author

Major comments

I think this manuscript describes timely and important experiments that, among other things, explicitly test the assumption that increased glucocorticoids are involved in moderating oxidative stress in response to an acute stressor. However, I think the evidence could be stronger, and suggest that the authors consider two points.

1. The authors feed CORT- or oil-injected mealworms to the quails and find that animals that ate CORT worms had significantly elevated CORT and ROMs (Experiment B). From this, they suggest that "GCs are at least partially responsible for the acute stress effect on oxidative stress". While I don't disagree, I think that to make this claim it is equally important to show the opposite, i.e. that blocking CORT during an acute stress response does not lead to an increase in oxidative stress parameters.

2. The authors show that acutely stressing animals leads to an increase in GC response and measures of oxidative stress (Experiment A). This is interesting but to me it lacks a negative control to convincingly rule out that any changes in ROMs/TAC levels are due to the stressor and not other factors (such as changes in GCs and other physiological processes that are under the influence of daily rhythms, etc). I am not necessarily saying that more experiments are needed - perhaps someone has already addressed this issue in the past?

Minor comments

Line 84 - I think it needs to be made clearer here why the authors were interested in testing 2 breeds of quail.

Line 233 - Could you test to see if the variation in response is larger in one of the breeds? This would be interesting to see if domestication/selection for growth characteristics (and perhaps other behavioural characteristics) also inadvertently led to selection of CORT responses to acute stress.

Line 264-265 - I think it's premature to suggest this, as finding the same thing in 2 different breeds of the same species does not make a strong case for generalisation.

Figure 1 - maybe change axes of B and C so we can see the change over time more clearly (it looks very modest, despite statistical significance)

Figure 2A - maybe put a box around the legend, the symbols look like data points. Also, from the figure it seems like the controls were consistently sampled a few minutes earlier than CORT fed. Is that just a glitch in the graph? It's also the case in fig 3a and 3b.

Decision letter (RSPB-2019-1698.R0)

23-Sep-2019

Dear Dr Hausmann:

Your manuscript has now been peer reviewed and the reviews have been assessed by an Associate Editor. The reviewers' comments (not including confidential comments to the Editor) and the comments from the Associate Editor are included at the end of this email for your reference. As you will see, the reviewers and the Editors have raised some concerns with your manuscript and we would like to invite you to revise your manuscript to address them.

Research ethics:

Use of animals and field studies:

It is a condition of publication that you make available the data and research materials supporting the results in the article. Datasets should be deposited in an appropriate publicly available repository and details of the associated accession number, link or DOI to the datasets must be included in the Data Accessibility section of the article

(<https://royalsociety.org/journals/ethics-policies/data-sharing-mining/>). Reference(s) to datasets should also be included in the reference list of the article with DOIs (where available).

Please submit a copy of your revised paper within three weeks. If we do not hear from you within this time your manuscript will be rejected. If you are unable to meet this deadline please let us know as soon as possible, as we may be able to grant a short extension.

Best wishes,
Dr Sasha Dall
mailto: proceedingsb@royalsociety.org

Associate Editor
Board Member: 1
Comments to Author:

I agree with both referees that this is a highly valuable study establishing a causal link between glucocorticoid release and oxidative stress. Both referees make valuable suggestions how to further improve the manuscript and I may add a stylistic one. I am fully aware that in the "hormone world" GC and CORT are typical acronyms for "glucocorticoids" and "corticosterone". However, I never understood why these acronyms are needed. Since your paper is already full of other acronyms that may be more useful because they stand for longer expressions (ROMs or GiOS I would suggest to spell out GCs and CORT, keeping the general readership of Proc. B in mind.

Sincerely
Wolfgang Goymann

Reviewer(s)' Comments to Author:

Referee: 1

Comments to the Author(s)

I thoroughly enjoyed reading this manuscript, which presents the most comprehensive test of the relationship between acute stress/glucocorticoids and oxidative stress to date. As such it provides an important and widely relevant advance on a topic that has received much speculation, but on which published data are relatively rare. I found it to be well-written and concise, and the findings are clear. I expect it to be of broad general interest.

My suggestions primarily consist of recommendations to better emphasize and provide context for some of the key findings.

First, the finding that moderate levels of prior stress decrease the ROMs response to subsequent stressors – but not the CORT response – is very interesting. These patterns, as well as the implication that moderate stressors may have hormetic effects on subsequent damage resulting from stressors, were well-described. But there wasn't much reference to the apparently paradoxical finding while CORT increases ROMs, moderate stress doesn't reduce the CORT stress response, or alter antioxidant capacity (at least as measured via TAC), but it does reduce the resulting oxidative damage. Presumably this is because some unmeasured components of antioxidant capacity or other physiological mediators increase to combat the direct effects of CORT on oxidative damage. The possibility of upregulating antioxidant defenses is mentioned briefly on lines 328-331, but the contrast between this and the finding of no change in TAC following repeated stressor exposure isn't explicitly discussed. Further discussion of what this might constitute given the lack of a response in measured antioxidants would strengthen the manuscript.

Similarly, noting that these results show that despite clear causal relationships between CORT and ROMs higher stress-induced CORT doesn't always equate to higher ROMs (as shown in the prior stressor experiment) would be helpful. Along those lines, can you say anything about why previous studies might not have found an increase in ROMs under acute stress?

The experimental design is elegant and well-justified, and it's clear that the birds need to be housed in pairs. Nevertheless I found myself wondering whether there was any relationship between dominance status in the quail pairs and the propensity to consume mealworms. It seems like the design of the experiment would minimize the risk that any such differences would affect the findings, since both animals ultimately consumed a mealworm in the actual experiment. But if there were differences in the perceived quality of CORT and control-injected mealworms this could still matter. Providing a very brief mention of whether there was any indication that individuals discriminated between mealworm types, and/or whether there was a relationship between dominance/the propensity to consume mealworms during the training phase and the type of worm that individuals consumed during the experimental phase would address this.

Other suggestions:

- Remove the word "them" in line 164

Referee: 2

Comments to the Author(s)

Major comments

I think this manuscript describes timely and important experiments that, among other things, explicitly test the assumption that increased glucocorticoids are involved in moderating oxidative stress in response to an acute stressor. However, I think the evidence could be stronger, and suggest that the authors consider two points.

1. The authors feed CORT- or oil-injected mealworms to the quails and find that animals that ate CORT worms had significantly elevated CORT and ROMs (Experiment B). From this, they suggest that "GCs are at least partially responsible for the acute stress effect on oxidative stress". While I don't disagree, I think that to make this claim it is equally important to show the opposite, i.e. that blocking CORT during an acute stress response does not lead to an increase in oxidative stress parameters.

2. The authors show that acutely stressing animals leads to an increase in GC response and measures of oxidative stress (Experiment A). This is interesting but to me it lacks a negative control to convincingly rule out that any changes in ROMs/TAC levels are due to the stressor and not other factors (such as changes in GCs and other physiological processes that are under the influence of daily rhythms, etc). I am not necessarily saying that more experiments are needed - perhaps someone has already addressed this issue in the past?

Minor comments

Line 84 - I think it needs to be made clearer here why the authors were interested in testing 2 breeds of quail.

Line 233 - Could you test to see if the variation in response is larger in one of the breeds? This would be interesting to see if domestication/selection for growth characteristics (and perhaps other behavioural characteristics) also inadvertently led to selection of CORT responses to acute stress.

Line 264-265 - I think it's premature to suggest this, as finding the same thing in 2 different breeds of the same species does not make a strong case for generalisation.

Figure 1 - maybe change axes of B and C so we can see the change over time more clearly (it looks very modest, despite statistical significance)

Figure 2A - maybe put a box around the legend, the symbols look like data points. Also, from the figure it seems like the controls were consistently sampled a few minutes earlier than CORT fed. Is that just a glitch in the graph? It's also the case in fig 3a and 3b.

Author's Response to Decision Letter for (RSPB-2019-1698.R0)

See Appendix A.

Decision letter (RSPB-2019-1698.R1)

21-Oct-2019

Dear Dr Hausmann

I am pleased to inform you that your manuscript entitled "Is there an oxidative cost of acute stress? Characterization, implication of glucocorticoids, and modulation by prior stress experience" has been accepted for publication in Proceedings B.

Open Access

You are invited to opt for Open Access, making your article freely available to all as soon as it is ready for publication under a CCBY licence. Our article processing charge for Open Access is £1700.

Paper charges

Sincerely,

Dr Sasha Dall

Appendix A

Revision of Manuscript ID RSBL-2013-0684

We have made the suggested changes by the referees and are we think the paper is much improved. We would like to thank the editor and both of the referees for their helpful comments during the review process.

Associate Editor

Board Member: 1

Comments to Author:

I agree with both referees that this is a highly valuable study establishing a causal link between glucocorticoid release and oxidative stress. Both referees make valuable suggestions how to further improve the manuscript and I may add a stylistic one. I am fully aware that in the "hormone world" GC and CORT are typical acronyms for "glucocorticoids" and "corticosterone". However, I never understood why these acronyms are needed. Since your paper is already full of other acronyms that may be more useful because they stand for longer expressions (ROMs or GiOS I would suggest to spell out GCs and CORT, keeping the general readership of Proc. B in mind.

Sincerely

Wolfgang Goymann

Thank you. We are excited about this work. We address each comment below (line number). We have also replaced the acronyms "GCs" and "CORT" with the words "glucocorticoids" and "corticosterone."

Referee: 1

Comments to the Author(s)

I thoroughly enjoyed reading this manuscript, which presents the most comprehensive test of the relationship between acute stress/glucocorticoids and oxidative stress to date. As such it provides an important and widely relevant advance on a topic that has received much speculation, but on which published data are relatively rare. I found it to be well-written and concise, and the findings are clear. I expect it to be of broad general interest.

We thank the referee for their general comments, and we share their enthusiasm about our data which explores the relationship between glucocorticoids and oxidative stress.

My suggestions primarily consist of recommendations to better emphasize and provide context for some of the key findings.

First, the finding that moderate levels of prior stress decrease the ROMs response to subsequent stressors – but not the CORT response – is very interesting. These patterns, as well as the implication that moderate stressors may have hormetic effects on subsequent damage resulting from stressors, were well-described. But there wasn't much reference to the apparently paradoxical finding while CORT increases ROMs, moderate stress doesn't reduce the CORT stress response, or alter antioxidant capacity (at least as measured via TAC), but it does reduce the resulting oxidative damage. Presumably this is because some unmeasured components of antioxidant capacity or other physiological mediators increase to combat the direct effects of CORT on oxidative damage. The possibility of upregulating antioxidant defenses is mentioned briefly on lines 328-331, but the contrast

between this and the finding of no change in TAC following repeated stressor exposure isn't explicitly discussed. Further discussion of what this might constitute given the lack of a response in measured antioxidants would strengthen the manuscript.

We thank the reviewer for their suggestions and have added a few sentences to the discussion elaborating on how antioxidant defenses other than those measured by TAC may change in response to stress exposure. Specifically, we discuss a recent study in penguins that found glutathione upregulation following an acute stressor (Stier et al., 2019). In this study, glutathione was also positively correlated with baseline corticosterone levels. We now summarize this study and suggest the possibility that prior stress exposure may alter specific antioxidant concentrations (lines 407-411).

Similarly, noting that these results show that despite clear causal relationships between CORT and ROMs higher stress-induced CORT doesn't always equate to higher ROMs (as shown in the prior stressor experiment) would be helpful. Along those lines, can you say anything about why previous studies might not have found an increase in ROMs under acute stress?

Previous studies are mixed, and this could partly be explained by the fact that prior experience with mild stressors modulates GiOS (exp C). In animals from natural populations, there is no way to know their prior experience with stressors, so previous exposure may explain in part the discrepancies in previous studies. One advantage to our lab controlled work is that we know that there was no prior stress exposure in the study. We have now included some discussion of this in the conclusion of the study (line 484-487).

The experimental design is elegant and well-justified, and it's clear that the birds need to be housed in pairs. Nevertheless I found myself wondering whether there was any relationship between dominance status in the quail pairs and the propensity to consume mealworms. It seems like the design of the experiment would minimize the risk that any such differences would affect the findings, since both animals ultimately consumed a mealworm in the actual experiment. But if there were differences in the perceived quality of CORT and control-injected mealworms this could still matter. Providing a very brief mention of whether there was any indication that individuals discriminated between mealworm types, and/or whether there was a relationship between dominance/the propensity to consume mealworms during the training phase and the type of worm that individuals consumed during the experimental phase would address this.

We also were curious about how social dominance may affect the feeding behavior of our quail, so we directly observed and videotaped all trials. The pens were large enough that aggressive interactions were extremely rare after the quail were habituated to the housing conditions. When the metal cover was raised, the quail that was closest to the mealworm was the bird who ate the mealworm. The bird that was closest to the plate varied from trial to trial, so it did not appear to us that one quail spent more time by the plate and got the worm more quickly. Due to the length of the manuscript and because we did not see any effect of dominance, we did not include these observations in the manuscript. However, based on the referee's question and our own interest in this possibility we have now included a short sentence about dominance behaviors in the method (line 154-156).

Other suggestions:

- Remove the word “them” in line 164

We have removed this word from line 164.

Referee: 2

Comments to the Author(s)

Major comments

I think this manuscript describes timely and important experiments that, among other things, explicitly test the assumption that increased glucocorticoids are involved in moderating oxidative stress in response to an acute stressor. However, I think the evidence could be stronger, and suggest that the authors consider two points.

1. The authors feed CORT- or oil-injected mealworms to the quails and find that animals that ate CORT worms had significantly elevated CORT and ROMs (Experiment B). From this, they suggest that “GCs are at least partially responsible for the acute stress effect on oxidative stress”. While I don’t disagree, I think that to make this claim it is equally important to show the opposite, i.e. that blocking CORT during an acute stress response does not lead to an increase in oxidative stress parameters.

We agree that such an experiment would provide useful insight into the relationship between glucocorticoids and oxidative stress and would strengthen the results of our study by testing if the absence of CORT stress response would be sufficient to prevent GiOS. However, it would be difficult to conduct such an experiment because blocking corticosterone would also affect the production of other hormones and would therefore likely to introduce confounding effects. However, we have added a sentence stating that future work could consider loss of function experiments (line 382-383).

2. The authors show that acutely stressing animals leads to an increase in GC response and measures of oxidative stress (Experiment A). This is interesting but to me it lacks a negative control to convincingly rule out that any changes in ROMs/TAC levels are due to the stressor and not other factors (such as changes in GCs and other physiological processes that are under the influence of daily rhythms, etc). I am not necessarily saying that more experiments are needed – perhaps someone has already addressed this issue in the past?

We agree that negative controls are important, which is why in Experiment B we tested the effects of acute increases of corticosterone in the absence of external stress. The negative control in this experiment was the quail who ate the mealworm with oil only. Because there was no increase in oxidative stress in these birds, it suggests that in this experiment the increase in oxidative stress was due to the corticosterone.

In experiment A, there isn’t a simple way to include a negative control, because taking blood samples (to measure CORT or oxidative stress) in and of itself is stressful. While we could measure a group of birds only at time 45 and compare those oxidative stress values to a group sampled at time 0 this would make the study cross-sectional vs. longitudinal and a loss of statistical power. When designing the experiment, we felt that within-individual measurements

were important because between-individual differences are already large at baseline. Thus, measuring within-individual changes is more powerful to detect GiOS. In addition, while the referee mentions processes that are under daily rhythms, differences in corticosterone or oxidative stress that are linked to daily rhythms are unlikely to occur in less than an hour.

Minor comments

Line 84 – I think it needs to be made clearer here why the authors were interested in testing 2 breeds of quail.

We added that we were interested in testing two phenotypically distinct breeds of quail to test whether changes in oxidative stress following an acute stressor differ in different organisms (line 97). In addition, we include that the feral quail have undergone less selection than domestic breeds which could affect GiOS (line 129-133).

Line 233 - Could you test to see if the variation in response is larger in one of the breeds? This would be interesting to see if domestication/selection for growth characteristics (and perhaps other behavioural characteristics) also inadvertently led to selection of CORT responses to acute stress.

While we agree with the referee that the variation in GCs between breeds is interesting, we are concerned about spending too much time trying to explain away a negative result. That said, we did point out to the reader that many of the domestic quail had modest increase in CORT. Based on this comment, we also analyzed deltaROM (Rom t45 – ROMt0) and deltaCORT (CORT t20-CORTt0) and there is not a significant difference. At this point, we don't think further post-hoc analyses are appropriate. However, we do appreciate the referees curiosity, and because the data will be made available, other interested scientists can do this type of exploratory analysis if they wish.

Line 264-265 – I think it's premature to suggest this, as finding the same thing in 2 different breeds of the same species does not make a strong case for generalisation.

We agree with the referee that it is premature to conclude that an increase in oxidative stress is a general cost of mounting an acute stress response. That is why we used the phrases 'suggesting' and 'may be a general cost' (line 327-329) and why we say 'We therefore need more research across diverse vertebrate taxa to determine the shared nature of the stress response and oxidative stress in order to better understand the extent to which GiOS is conserved.' (line 352-354).

Figure 1 - maybe change axes of B and C so we can see the change over time more clearly (it looks very modest, despite statistical significance)

We agree that, while significant, the change over time is not large. We're not sure how changing the axes will address this, and think it is instructive for the reader to see these small, though significant effects.

Figure 2A - maybe put a box around the legend, the symbols look like data points. Also, from the figure it seems like the controls were consistently sampled a few minutes earlier than CORT fed. Is that just a glitch in the graph? It's also the case in fig 3a and 3b.

just mention in the legend that groups are slightly separated on the x-axis just for clarity / visual purpose.

We have added that points on the scatterplots were jittered to the figure captions for Figure 2 and Figure 3. We have also added boxes to the figure legend 2A.